

# Pathological game use in adults with and without Autism Spectrum Disorder

Christopher R. Engelhardt[1], Micah O. Mazurek[2,3] and Joseph Hilgard[4]

[1] CARFAX, Inc., Columbia, MO, United States of America
[2] Department of Health Psychology and Thompson Center for Autism and Neurodevelopmental Disorders, University of Missouri, Columbia, MO, United States of America
[3] Curry School of Education, University of Virginia, Charlottesville, VA, United States of America
[4] Annenberg Public Policy Center, University of Pennsylvania, Philadelphia, PA, United States of America

## ABSTRACT

This study tested whether adults with autism spectrum disorder (ASD) are at higher risk for pathological game use than typically developing (TD) adults. Participants included 119 adults with and without ASD. Participants completed measures assessing daily hours of video game use, percent of free time spent playing video games, and symptoms of pathological game use. The results indicated that adults with ASD endorsed more symptoms of video game pathology than did TD adults. This relationship was strong, enjoying 300,000-to-1 odds in Bayesian model comparison. Results also showed that adults with ASD spent more daily hours playing video games and spent a higher percent of their free time playing video games than did TD adults. Even after adjustment for these differences in daily video game hours and proportion of free time spent on games, model comparisons found evidence for a difference in game pathology scores associated with ASD status. Additionally, escapism motives for playing video games was associated with game pathology scores in both ASD and TD adults, replicating and extending a previous report. In conclusion, the risk for pathological game use appears larger in adults with ASD compared with TD adults. These findings point to pathological game use as a potentially important focus of clinical attention in adults with ASD.

Corresponding authors
Micah O. Mazurek,
mazurekm@health.missouri.edu,
mazurekm@virginia.edu
Joseph Hilgard, jhilgard@gmail.com

## INTRODUCTION

Autism spectrum disorder (ASD) is characterized by core deficits in social interaction/communication and by restricted interests and repetitive behaviors (*American Psychiatric Association, 2013*). These core impairments, which are often accompanied by co-occurring mental health problems such as depression and anxiety (*Gillott & Standen, 2007*; *Mazefsky, Folstein & Lainhart, 2008*; *Moseley et al., 2011*), persist through adulthood (*Seltzer et al., 2004*; *Shattuck et al., 2007*; *Taylor & Seltzer, 2010*). As a result, adults with ASD have limited engagement in social and community activities and are at risk for poor independent living and employment outcomes (*Eaves & Ho, 2008*; *Howlin et al., 2004*; *Orsmond & Kuo, 2011*). Thus, identifying factors that affect health, well-being, and daily functioning in adults with ASD is of chief importance.

Among these factors, excessive video game play may be an important consideration. Because social interactions are often hyper-stimulating or anxiety-provoking for individuals with ASD (*Dalton et al., 2005*; *Joseph et al., 2008*), video games may be an especially appealing activity due to their predictability and low social demands. Consistent with this line of reasoning, previous studies have found that children and adolescents with ASD spent more than two hours per day playing video games. The time spent playing video games is greater than that of children with other disabilities (e.g., speech/language impairments, intellectual disabilities) or that of typically developing (TD) children (*Mazurek & Engelhardt, 2013b*; *Mazurek et al., 2012*). Children with ASD also play more video games than do their TD siblings and spend far more time playing video games than on other extracurricular activities (*Mazurek & Wenstrup, 2013*). Excessive play may be deleterious; problematic game use among children with ASD was associated with poor sleep and poorer behavioral functioning (*Engelhardt, Mazurek & Sohl, 2013*; *Mazurek & Engelhardt, 2013a*). These findings are consistent with anecdotal reports from parents and clinicians that many individuals with ASD exhibit patterns of excessive video game play.

## Pathological video game use

Pathological video game use is a significant clinical issue in its own right, included as "Internet Gaming Disorder", a "Condition for Further Study", in the DSM-5 (*American Psychiatric Association, 2013*). Although broad consensus has not been reached, many researchers view the construct as similar to other behavioral addictions such as gambling addiction (*Griffiths, 2008*; *Griffiths, Kuss & King, 2012*; *Hellman et al., 2013*; *King et al., 2013*). Like gambling, video game play can be a source of enjoyment. However, video game play can become pathological if it leads to significant impairment or distress. Consistent with the definition of internet gaming disorder and gambling disorder in the DSM-5 (*American Psychiatric Association, 2013*), pathological game use is defined by its symptoms: preoccupation with video games, salience, euphoria/relief, tolerance, withdrawal symptoms, mood modification, conflict, relapse and reinstatement, unsuccessful attempts to control video game play, continued excessive use, deceiving others as to the amount of time spent gaming, and loss of interest in hobbies (*American Psychiatric Association, 2013*; *King et al., 2013*; *Petry et al., 2014*; *Sim et al., 2012*). Previous research has found correlations between symptoms and amount of video game play, video game play to escape real life, self-perceptions of excessive game play, and scores on other measures of dependence (*Gentile, 2009*; *Hart et al., 2009*; *Hilgard, Engelhardt & Bartholow, 2013*; *King, Delfabbro & Zajac, 2011*; *King et al., 2013*; *Lemmens, Valkenburg & Peter, 2009*; *Sim et al., 2012*; *Tejeiro Salguero & Morán, 2002*). This research supports pathological game use as a reliable and valid construct (but see *Scharkow, Festl & Quandt (2014)* for concerns about test-retest stability and reliability).

Pathological game use is a clinically relevant problem in the general population. Adults who play video games to excess have less time for other activities, including work, educational pursuits, hobbies, socialization with friends and family members, and sleep (*Eickhoff et al., 2015*; *Griffiths, Davies & Chappell, 2003*). In addition, pathological video game use is associated with a number of negative psychosocial indicators, including

aggression (*Griffiths & Hunt, 1998*; *Grüsser, Thalemann & Griffiths, 2006*), depression (*Feng et al., 2003*; *Gentile et al., 2011*), anxiety (*Gentile et al., 2011*), poor life satisfaction (*Lemmens, Valkenburg & Peter, 2011*; *Mentzoni et al., 2011*), loneliness (*Lemmens, Valkenburg & Peter, 2009*), and interference with social relationships (*Lemmens, Valkenburg & Peter, 2011*; *Smyth, 2007*). Negative health effects also include reduced sleep (*Smyth, 2007*), risk of game-induced seizures (*Chuang, 2006*), and poor physical health (*Kelley & Gruber, 2012*). Overall, both researchers and clinicians have concluded that the real-world problems associated with pathological game use are worthy of future investigation.

The well-being of adults with ASD could benefit from attention to potential symptoms of pathological game use. Individuals with ASD are at risk for social isolation, preoccupation with specific interests, and repetitive patterns of behavior (*American Psychiatric Association, 2013*). As such, individuals with ASD may be at particular risk for excessive use of video games.

Consistent with this idea, prior studies have found that individuals with ASD may have a complicated relationship with video games. On the one hand, qualitative findings suggest that adults with ASD report a number of positive aspects of video game use, including stress reduction and social connection (*Mazurek, Engelhardt & Clark, 2015*). On the other hand, children and adolescents with ASD have difficulty disengaging from video games (*MacMullin, Lunsky & Weiss, 2015*; *Mazurek & Engelhardt, 2013b*; *Mazurek & Wenstrup, 2013*). For example, research has shown that boys and girls with ASD endorsed more symptoms of pathological game use than did their TD siblings (*Mazurek & Wenstrup, 2013*). This group difference was replicated in a sample of unrelated boys with and without ASD, finding a large ($d$ >1) difference between groups (*Mazurek & Engelhardt, 2013b*). Results such as these indicate a clinically meaningful difference in pathological game use in individuals with and without ASD.

Although previous research has documented differences in pathological game use in *children and adolescents* with and without ASD, no study has examined whether *adults* with ASD are at higher risk for pathological game use than TD adults. This is a particularly important research topic because adults with ASD often suffer from poor social outcomes (*Billstedt, Gillberg & Gillberg, 2005*; *Billstedt, Gillberg & Gillberg, 2007*; *Howlin et al., 2004*; *Levy & Perry, 2011*; *Müller, Schuler & Yates, 2008*). As adults have greater responsibilities and autonomy than children, preoccupation with video games in individuals with ASD may be more harmful to functioning among adults than among children. Difficulty disengaging from video games may prevent adults from engaging in productive employment, education, or community activities.

Consistent with this idea, recent qualitative research has shown that adults with ASD express concerns about their preoccupation with video games. In this research, 10–15% of participants reported playing video games out of compulsion or routine and voiced concerns about feeling addicted to games. Many also expressed that they played games in order to fill time and that they often played games to the exclusion of other activities (*Mazurek, Engelhardt & Clark, 2015*). Thus, there is reason to suspect that adults with ASD might be at greater risk for pathological game use than adults without ASD.

## Current study

The goal of the current study was to examine differences in video game play in adults with and without ASD. The primary prediction was that adults with ASD would be at higher risk for pathological game use than TD adults. We also predicted that adults with ASD would play more video games per day and spend a higher percent of their free time playing video games than TD adults. We tested these predictions using a model comparison approach with Bayes factors, enabling us to directly examine the amount of evidence in favor of or against a group difference in pathological game use.

## METHODS

This manuscript represents a secondary analysis of a dataset collected in a previous report (*Engelhardt et al., 2015*). In that publication, we reported the results of an experiment in which ASD and TD individuals were randomly assigned to play a violent or nonviolent game; the effects of game violence on aggressive thoughts, feelings, and behaviors were described. In this publication, we describe correlations in questionnaire data collected alongside that experiment. A more comprehensive description of participant demographics, eligibility criteria, recruitment, and study procedure can be found in the original report.

The study was approved by our institutional review board (University of Missouri approval number #1206273). All participants provided written consent prior to participating in the study. Participants received $20 in exchange for their participation. Data and materials are available at https://osf.io/5dsnc/.

## Participants

Participants included 119 adults (16 women), half of whom had a previous diagnosis of ASD, participating in a larger study on violent video game effects (*Engelhardt et al., 2015*). One individual with ASD participated in the primary study but elected to withdraw from it prior to completing the measures in this cross-sectional study. The sample was primarily White (85% of the ASD group, 93% of the TD group) and non-Hispanic (85% of the ASD group, 100% of the TD group).

Participants in the ASD group were recruited through an academic medical center specializing in ASD diagnosis and treatment. These participants had a previous diagnosis of ASD based on the center's clinical care model. The diagnostic process generally includes structured interviews, behavioral observation, and evaluations conducted by psychologists and/or physicians using standardized assessment tools such as the Autism Diagnostic Observation Schedule (ADOS) (*Lord, DiLavorne & Risi, 2002*) and/or the Autism Diagnostic Interview—Revised (ADI-R) (*Lord, Rutter & Le Couteur, 1994*). All participants in the ASD group were verbally fluent and high-functioning, with IQ scores greater than or equal to 85. Most participants in the ASD group were attending school (39% full-time and 20% part-time). The majority (64%) of participants in the ASD group were not currently employed, although 3% had full-time jobs and 32% had part-time jobs.

Participants in the TD group were recruited through the community and the university campus using email, campus flyers, and face-to-face recruitment. Participants in the TD group had no history of neurological or developmental disorders. Most participants in the
**Table 1** Demographic Characteristics of the Two Diagnostic Groups.

| Group | Male (%) | Age | ABIQ | AQ | Daily gameplay (hrs) Weekday | Weekend | Escapism |
|---|---|---|---|---|---|---|---|
| ASD | 86 | 20.42 | 103.1 | 70.24 | 2.49 | 3.57 | 3.61 |
| TD | 87 | 20.54 | 103.4 | 57.17 | 0.83 | 1.56 | 2.89 |

Notes.

ABIQ, Abbreviated Battery IQ; AQ, Autism Short Questionnaire; Escapism, Escapism subscale score from GAMES questionnaire.

TD group were also attending school (95% full-time and 3% part-time), and most were also currently employed (3% full-time and 65% part-time). The ASD and TD groups did not differ in age, gender, or IQ. Demographics are supplied in Table 1. Further details are available in *Engelhardt et al. (2015)*

## Measures
### Video game use
Participants reported the number of hours over the past year they spent playing video games on a typical weekday and weekend day. These were highly correlated ($r \geq .85$ in both groups), so they were combined to make average time spent playing video games per day, calculated by multiplying the weekday value by 5 and the weekend value by 2, summing these scores, then dividing by 7, consistent with previous research (*Orsmond & Kuo, 2011*). Participants also indicated the percent of their free time spent playing video games. At the end of the study, the researcher asked "What percent of your free time do you spend playing video games?" If the respondent answered with a range of values, the midpoint was used for analysis (e.g., "15–20%" was coded as 17.5%). Although we expect these two variables to be positively correlated, we do not consider them as the same construct: depending on one's obligations, a 1-hour game session could occupy a little of one's free time or all of it.

### Pathological game use
Participants completed a 10-item measure of pathological video game use adapted from previous research (*Gentile, 2009*; *Hilgard, Engelhardt & Bartholow, 2013*). The measure was originally developed to be consistent with DSM-IV criteria for another behavioral addiction (pathological gambling), and items shared features with widely-used definitions of addiction, including salience, euphoria or relief, tolerance, withdrawal, conflict, relapse and reinstatement (see for detail *Gentile, 2009*). For example, the measure includes questions about withdrawal ("In the past year, have you become restless or irritable when attempting to cut down or stop playing video games?"), conflict with health and hygiene ("In the past year, have you ever skipped sleeping, eating, or bathing so that you could spend more time playing video games?"), preoccupation ("In the past year, have you become more preoccupied with playing video games, studying video game playing, or planning the next chance to play?"), tolerance ("In the past year, have you needed to spend more and more time on video games to achieve the same level of excitement?"), and deceit ("In the past year, have you ever lied to family or friends about how much you

play video games?"). Participants indicated whether they had experienced each symptom over the previous year by responding "*Yes*," "*Sometimes*," or "*No*," scored as 1, 0.5, and 0 points, respectively (*Gentile, 2009*; *Hilgard, Engelhardt & Bartholow, 2013*). The scale has demonstrated good convergent and divergent construct validity in prior research (*Gentile, 2009*). Cronbach's alpha coefficients were used to evaluate internal consistency of the scale in the current sample. Item reliability estimates were good, with Cronbach's alpha of .788 for the scale for the entire sample, $\alpha = .720$ for the ASD group, and $\alpha = .810$ for the TD group.

### Escapism motives

As a secondary analysis, we hoped to replicate the relationship between escapism motives for game use and symptoms of PGU we had found in an earlier report (*Hilgard, Engelhardt & Bartholow, 2013*). Participants rated each item from the GAMES escapism subscale on a 1–5 scale (*Hilgard, Engelhardt & Bartholow, 2013*).

# RESULTS

## Analytic approach

In response to numerous critiques levied against null hypothesis significance tests and 95% confidence intervals, we used Bayes factors to state evidence for or against our predictions. Bayes factors are reported in ratios such as 4-to-1 in favor of or against statistical models. These evidence ratios can be interpreted as the probability of the observed data under one model compared with the probability of the observed data under a different model. They can also be interpreted as how beliefs should be updated in light of data. Computations were conducted in the BayesFactor package for **R** (*Morey, Rouder & Jamil, 2014*) using the GeneralTestBF function and 100,000 Monte Carlo iterations.

We examined evidence for all possible model combinations. This strategy enabled us to determine the best-fitting model and the amount of evidence in favor of or against a parameter in that model. For example, in the analysis in which game pathology scores were included as the dependent variable and group diagnosis, average time spent playing video games per day, and the percent of free time spent playing video games were included as the independent variables, seven models were compared: the null effects model, the model with group diagnosis only, the model with average time spent playing video games per day only, the model with percent of free time spent playing video games only, three models including two of the three parameters, and a full model including all three parameters.

The models presented below used a JZS default prior. This Cauchy-distributed prior has spread to cover reasonable effect sizes, specifies that smaller effects are more likely than larger effects, and is centered at 0. Consistent with previous research showing that boys with ASD experience more problems with problematic video game use compared with TD boys (*Mazurek & Engelhardt, 2013b*), we tuned the scale parameter for the prior on the effects of both categorical and continuous predictors to 1.12, a large effect. This prior assigns 50% of probability to effects $|\delta| = 1.12$ or smaller. The remaining probability is assigned to larger effect sizes.

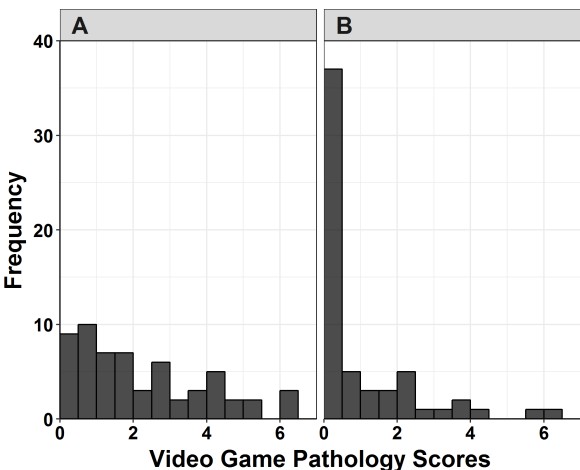

**Figure 1** **Histograms of pathological game use scores by diagnostic group.** (A) Histogram of scores in ASD group. (B) Histogram of scores in TD group.

We used general linear models to test our predictions. Because visual inspection of the pathological game use scores indicated positive skew (Fig. 1), a Box–Cox transformation was applied to this outcome. The suggested transformation, raising game-pathology scores to the $-.26$ power, helped normalize the residuals. We analyzed this transformed variable in the general linear models presented below.

## Symptoms of pathological game use

Our primary prediction was that adults with ASD would exhibit higher pathological game use scores than TD adults. Adults with ASD indicated more symptoms than did TD adults (untransformed $M(SD) = 2.46\ (1.77)$ and $1.20\ (1.55)$ for the ASD and TD groups, respectively, $d = 0.84\ [0.47, 1.22]$, $t(117) = 4.61$. Bayesian analysis indicated that the observed difference of transformed values was 300,000 times more likely to reflect a hypothesized difference than no difference.

We next explored which features served to predict pathological game use scores through the comparison of multiple regression models. These models could include as predictors each of group diagnosis, average time spent playing video games per day, and/or the percent of free time spent playing video games. This lead to seven candidate models, representing the inclusion of one, two, or all three additive effects. All seven models were preferred to the null effects model. Bayes factor model comparisons suggested that the best-fitting model included parameters for diagnostic group ($b = 0.22$) and for the percent of free time spent playing video games ($b = 0.01$). This model was strongly preferred to the null effects model by a factor of $6.4 \times 10^{11}$ (or in frequentist terms, $F(2, 116) = 42.1$).

By dropping individual terms from this model, one can assess the evidence for each effect conditional on adjustment for the other effects. For example, we dropped the effect of autism group from the above model, comparing the change in model fit with and without this effect. This model comparison yielded a Bayes factor of 3.0-to-1 ($t(116) = 2.61$) in favor of an effect for diagnostic group, even after adjustment for the effect of percentage of free time spent playing video games. Similar model comparisons yielded a Bayes factor of

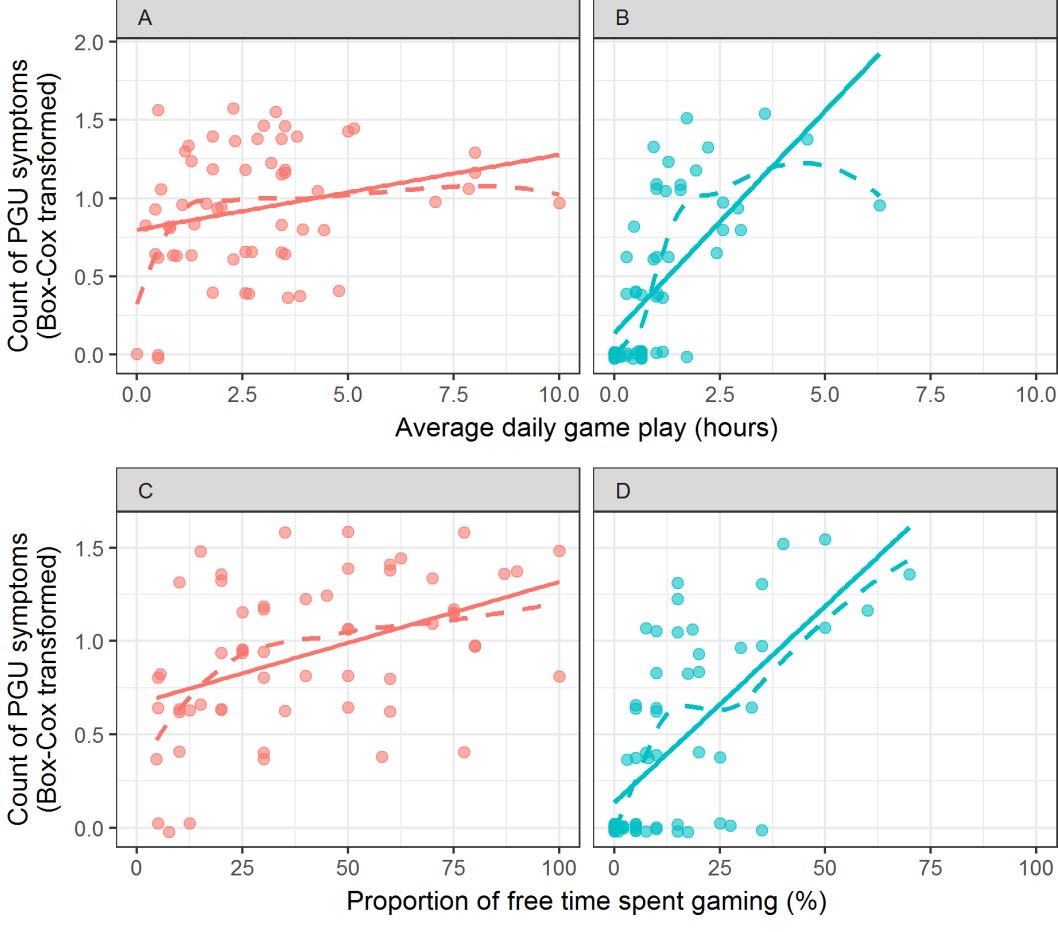

**Figure 2  Relationships between time spent gaming (hours, (A) and (B), and proportion, (C) and (D)) and PGU symptom count (Box–Cox transformed).** Relationships are plotted separately for ASD (A and C) and TD (B and D) groups. Slight vertical jitter has been added to datapoints to reduce overplotting. Linear regression displayed in solid line, locally weighted average displayed in dashed line.

more than 1,400,000-to-1 ($t(116) = 6.18$) in favor of an effect for the percent of free time spent playing video games, even after adjustment for the effect of diagnostic group.

These regression models were then performed separately within each group to explore how their influence might differ between ASD and TD adults. Scatterplots are provided in Fig. 2. Within the ASD group, the best-fitting model modeled symptoms as a function of the proportion of free time, $F(1, 57) = 13.5$, $BF = 37$, $b = 0.01$. Average daily game play did not strongly predict symptoms whether adjusting for ($BF = 0.14$, or 7.3-to-1 against, $b = 0.01$, $t(56) = 0.26$) or not adjusting for ($BF = 0.58$, or 1.71-to-1 against, $b = 0.05$, $t(57) = 1.97$) the proportion of free time spent playing games. Within the TD group, the best-fitting model modeled symptoms as a function of the additive effects of both proportion of free time ($b = 0.01$, $t(57) = 2.84$) and daily hours spent playing video games ($b = 0.17$, $t(57) = 3.07$), $BF = 1.3 \times 10^7$, $F(2, 57) = 30.75$. For this group, average daily game play did provide additional prediction of symptoms, even when adjusting for the proportion of free time spent playing games, $BF = 4.73$.
**Table 2  Bivariate relationships among all study variables.**

|  | Group | Gender | Hours | FreeTime% | Escapism | RawPGU | PGU |
|---|---|---|---|---|---|---|---|
| Group | 1.00 | | | | | | |
| Gender | .00 | 1.00 | | | | | |
| Daily game use (hrs) | .46 | .11 | 1.00 | | | | |
| Free time gaming (%) | .52 | .26 | .67 | 1.00 | | | |
| GAMES Escapism | .38 | .32 | .33 | .52 | 1.00 | | |
| PGU symptoms (raw) | .39 | .27 | .44 | .59 | .47 | 1.00 | |
| PGU symptoms (transformed) | .48 | .31 | .53 | .62 | .55 | .94 | 1.00 |

**Notes.**
Diagnostic Group was coded 0, typically developing; 1, autism spectrum disorder.
Gender was coded 0, female; 1, male.

**Table 3  Correlations per diagnostic group.**

|  | Gender | Hours | FreeTime% | Escapism | RawPGU | PGU |
|---|---|---|---|---|---|---|
| Gender | – | .05 | .31 | .27 | .33 | .39 |
| Daily game use (hrs) | .28 | – | .51 | .06 | .18 | .25 |
| Free time gaming (%) | .33 | .72 | – | .40 | .40 | .44 |
| GAMES Escapism | .42 | .42 | .48 | – | .34 | .40 |
| PGU symptoms (raw) | .27 | .62 | .69 | .42 | – | .94 |
| PGU symptoms (transformed) | .34 | .67 | .66 | .49 | .94 | – |

**Notes.**
Correlations calculated separately by group, with the ASD group displayed above the diagonal and the TD group displayed below the diagonal. Gender was coded 0, female; 1, male.

## Tests of second prediction

Zero-order correlations between all study variables can be seen in Table 2. Adults with ASD spent more time playing video games per day ($M = 2.80$ h, $SD = 2.14$) and spent a greater percent of their free time playing video games ($M = 41.02, SD = 27.38$) than did TD adults ($Ms = 1.04$ h (1.20) and 14.25 (15.94), respectively). These effects were large in magnitude, $d = 1.01$ [0.63, 1.40], $t(117) = 5.53$, and $d = 1.20$ [0.80, 1.59], $t(117) = 6.53$. Bayes factors found strong evidence of group differences in both of these variables: in each analysis, the model with group diagnosis was preferred to the null effects model by more than a factor of 59,000-to-1. Table 2 also shows that pathological game use symptoms are highly and positively associated with both average time spent playing video games per day and with the percent of free time spent playing video games, consistent with a number of previous studies. Table 3 presents the bivariate correlations for TD and ASD groups separately.

### Replication of Hilgard, Engelhardt & Bartholow (2013)

In a previous study, *Hilgard, Engelhardt & Bartholow (2013)* developed a questionnaire measure of motives for playing video games. In that report, they found a positive relationship between playing games to escape real-life problems and symptoms of PGU. We replicated this relationship in both the ASD ($b = 0.21$, $BF = 19.9$, $t(57) = 3.29$) and

TD ($b = 0.25$, $BF = 265$, $t(58) = 4.24$) groups. Additionally, escapism scores were significantly higher in the ASD group as compared to the TD group, $d = 0.81$ [0.43, 1.19], $BF = 846$, $t(117) = 4.42$.

## DISCUSSION

This is the first study to examine pathological game use in adults with and without ASD. Consistent with our primary prediction, we found very strong evidence that adults with ASD are at higher risk for pathological game use symptoms than TD adults. Evidence for this group difference remained even after modeling the average amount of time spent playing video games per day and the percent of free time spent playing video games. Consistent with our second prediction, we found very strong evidence that adults with ASD both spent more time playing video games per day and spent a greater percentage of their free time playing video games than TD adults. These findings are consistent with previous research among children with ASD (*Mazurek & Engelhardt, 2013b*; *Mazurek & Wenstrup, 2013*), suggesting that the greater risk for pathological game use may persist into adulthood for many individuals with ASD.

Additionally, some individual differences were found to correlate with PGU symptom counts. Among the TD group, both greater hours per day and greater proportion of free time spent gaming were associated with greater symptom counts. Among the ASD group, however, only proportion of free time spent gaming had a strong correlation with symptom counts. Relationships between these measures and symptom counts were weaker in the ASD group than in the TD group; whether this is because the ASD group has greater difficulty with self-report or less regimented use of time is unclear. Still, both groups showed a relationship between escapism motives and PGU symptoms, replicating and extending the result described in *Hilgard, Engelhardt & Bartholow (2013)*.

The results here suggest that pathological game use may be an important clinical consideration for this population. Adults with ASD may be at high risk for problematic game use due to particular aspects of the behavioral phenotype of ASD. For example, highly restricted interests, preoccupations, and perseverative interests are key diagnostic features of ASD (*American Psychiatric Association, 2013*). For some individuals, these behavioral features may manifest as preoccupations with video games or compulsive patterns of game-play. Problematic game-play patterns may themselves have additional deleterious effects on overall functioning. Adults with ASD are already at risk for poor outcomes, including reduced engagement in social and community activities, education, and employment (*Eaves & Ho, 2008*; *Howlin et al., 2004*; *Orsmond et al., 2013*). Excessive or pathological use of video games may exacerbate these difficulties, displacing time that could be spent on social, occupational, or other recreational activities (*Mazurek, Engelhardt & Clark, 2015*; *Mazurek & Wenstrup, 2013*). Furthermore, given evidence from the general population, pathological game use may exacerbate core or co-occurring symptoms among adults with ASD, particularly worsening mood, anxiety, irritability, and social isolation (*Feng et al., 2003*; *Gentile et al., 2011*; *Griffiths & Hunt, 1998*; *Grüsser, Thalemann & Griffiths, 2006*; *Lemmens, Valkenburg & Peter, 2011*; *Mentzoni et al., 2011*; *Smyth, 2007*). Additional

research is needed to examine the specific psychosocial consequences of pathological game use in adults with ASD. This information will be critical for informing effective interventions for this problem.

### Limitations and future directions

These data are cross-sectional and do not permit a causal interpretation of the current results. Longitudinal and experimental studies are necessary to determine both the predictors, mediators, and outcomes of pathological game use among individuals with ASD. Additionally, we did not measure the absolute amount of free time participants had. Individuals with ASD may have fewer responsibilities (e.g., school, employment) than TD peers, which may account in part for their greater use of video games. Whether pathological game use is a cause or consequence of these reduced responsibilities is beyond the scope of this study, but it may be interesting to research in longitudinal designs.

Another potential limitation of the study is its reliance on self-report measurements. This strategy could be problematic because previous studies have suggested that individuals with ASD have difficulty understanding and reporting on their own experiences (*Ben Shalom et al., 2006*; *Losh & Capps, 2006*). This may be why relationships between time spent playing games and PGU symptoms were weaker in the ASD group than the TD group. However, in order to enhance the validity of responses, participants were given an opportunity to ask the experimenter for clarification if survey questions seemed confusing. Participants were also pre-screened to ensure that they were fluent in English, could read and write on their own, and had an estimated IQ score greater than 85 prior to study enrollment.

Another potential limitation of the current study is that the measure of pathological video game use was relatively brief and covered a very wide time span (i.e., the past year). Although this measure has demonstrated good construct validity in prior research (*Gentile, 2009*), future work on this topic may benefit from further development of more comprehensive measures focused on a shorter recall period to enhance reliability and validity of responses.

Although the current study focused primarily on pathological aspects of video game use, there is also room for research into the potential positive aspects of game use among adults with ASD. Parents, researchers, and practitioners are interested in the potential of games for improving social and behavioral functioning in children with ASD (*Durkin et al., 2015*; *Ferguson et al., 2012*; *Grynszpan et al., 2014*; *Moore & Taylor, 2000*). Expanding this work to include a focus on improving outcomes for adults with ASD would be beneficial. Overall, more research is needed to inform the development of interventions focused both on positive applications of game-based technology and on reduction of potentially negative effects of pathological game use.

## CONCLUSIONS

These results suggest that adults with ASD may be at risk for pathological game use. Future research is needed to identify the prevalence, course, predictors, and outcomes of video game pathology in individuals with ASD. The current findings suggest that clinicians should

consider assessing these potential problems in clients. It may be important to consider the potential impact of video game use on emotional, behavioral, and social outcomes among adults with ASD.

### Abbreviations

| | |
|---|---|
| **ASD** | autism spectrum disorder |
| **DSM** | Diagnostic and Statistical Manual of Mental Disorders |
| **TD** | typically developing |
| *M* | mean |
| *SD* | standard deviation |

### Funding

This project was supported by the Richard Wallace Faculty Incentive Grant Award and the School of Health Professions Research Facilitation Fund at the University of Missouri. The funders had no role in study design, data collection and analysis, decision to publish, or preparation of the manuscript.

### Grant Disclosures

The following grant information was disclosed by the authors:
Richard Wallace Faculty Incentive Grant Award.
School of Health Professions Research Facilitation Fund at the University of Missouri.

### Competing Interests

The authors declare there are no competing interests. Christopher R. Engelhardt is an employee of CARFAX, Inc., Columbia, Missouri, United States.

### Author Contributions

- Christopher R. Engelhardt conceived and designed the experiments, performed the experiments, analyzed the data, wrote the paper, prepared figures and/or tables, reviewed drafts of the paper.
- Micah O. Mazurek conceived and designed the experiments, performed the experiments, wrote the paper, reviewed drafts of the paper.
- Joseph Hilgard conceived and designed the experiments, analyzed the data, wrote the paper, prepared figures and/or tables, reviewed drafts of the paper.

### Human Ethics

The following information was supplied relating to ethical approvals (i.e., approving body and any reference numbers):
The University of Missouri granted ethical approval to perform this study (Project # 1206273).

### Data Availability

OSF: https://osf.io/5dsnc/?view_only=70a90da31b914740b795d29d48f66af6.

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
