# Peer review of "Pathological game use in adults with and without Autism Spectrum Disorder"

_PeerJ, doi:10.7717/peerj.3393_

## Round 0.1 · original submission · Major Revisions

Thanks for your submission.

The reviewers raise a number of important concerns. I'm unsure how easy it will be for you to address these issues with the current dataset but I think it's important to give you an opportunity to respond.

In particular, Reviewer 1 notes that group differences in the gaming addiction measure may simply be a function of group differences in free time. It's really not clear to me that a control group of undergraduates (many of whom report having zero free time) is appropriate for comparison with autistic adults, the majority of whom are unemployed and potentially looking for activities to fill up their days.

I note that your analyses do show that group membership has an effect even after accounting for percentage of free time spent playing video games (line 262), but these kinds of analyses are problematic if the groups are already mismatched on what is essentially a covariate.

So I agree with Reviewer 1 that it is essential for the groups to be analysed separately in addition to any analysis that combines the two groups.

Given the concerns with the direct comparison, I wonder whether there is much to be gained by including the control group at all. In other words, it might be worth reframing the question as "What predicts problematic video game use amongst autistic adults?" Given the enormous heterogeneity within autism, there's potentially a lot of value in exploring individual differences within the diagnostic group.

Another possibility would be to compare autistic and control adults (from your current sample) who are matched for total amount of video game use. That would at least allow you to ask whether autistic adults are more prone to problematic video game use - and to see whether patterns of problematic game use across items are comparable across groups. Again, this is just a suggestion. I'm not sure how feasible this might be given the data you have.

A couple of other points I wanted to add to the reviewers' comments:

1. This is a "secondary analysis" of previously published research. It would be helpful to briefly summarise the findings of the primary analysis and clarify how the current analyses differ / are complementary.

2. At the end of the Discussion, you note that video game use may have benefits as well as negative outcomes. I think it is worth making this point early in the Introduction. (ie video game use has potential positives, but also there are concerns).

Reviewer 1 ·

Basic reporting

The text of the manuscript overall is clear, the raw data is supplied, and the study conforms to PeerJ standards. The topic is potentially of importance to the field, particularly to autism researchers.

Experimental design

In addition, the overall experimental design and methods overall are sound. However, there are several major concerns related to the methods and validity of findings which should be addressed.

1) For the individuals with zero hours of free time (11 TD individuals and no autistic individuals), the methods should discuss how proportions were created for these participants (0/0 is indeterminate mathematically). In addition, there were disproportionately more typically developing individuals who did not play video games at all (N = 12; primarily those with no free time) compared to the autism group (N = 1). As these individuals could not have reported pathological use (due to no use at all), inclusion of non-gamers in the sample will potentially skew the results towards lower representation of game addiction in the TD group than the autism group. The reasoning to include these participants and the analysis strategy used to handle responses of 0 should be discussed.

2) The groups do not appear to be matched on employment or schooling rates, but a comparison of these rates also do not appear to be included in either this study or the Englehardt et al. (2015) paper. The TD undergrads in particular seem to be very busy people with having both full time schooling and many also having part-time jobs at the same time, although employment data does not appear to be included in the raw data uploaded with the paper. The differences in schooling and employment rates may be due to differences in the recruitment methods, which might undermine the comparison of groups and should be addressed. The current paper should also include at least a short summary of the basic demographic information, such as the ages of each group included in the study, ethnicity, and recruitment methods.There does not need to be as much detail as was presented previously, but at least some basic information on demographics would be helpful to readers, as group matching strategies is an important topic in autism research (Jarold & Brock, 2004).

3) The authors should present models of the relation between game play (or proportion of free time) and video game addiction scores separately for each group, as this relationship would be expected to at least be present for each group given the sample sizes and previous literature cited. While implied, these results are never analyzed separately for the two groups. A quick analysis of the data provided suggests that for the autism group alone, the Pearson Correlation between average hours spent playing games and video game addiction appears to be non-significant, r = .18, p = .17. In fact, one of the autistic individuals with a very low amount of video game total play time (.5 hours per day) has a particularly high score for video game addiction (6.5), whereas the autistic individual with the highest amount of game time (10 hours per day) has a much lower video game addiction score (2). This suggests that the measure of video game addiction may not actually be a useful measure for individuals with ASD, even if it may be useful for other populations, and thus needs to be explored more fully prior to collapsing all ASD and TD participants into the same models.

Validity of the findings

Inspection of the data suggests to me that one reason there may be relatively more TD adults who don’t play games is that the groups are also not matched in terms of their total amount of free time. The autism group reported having on average 41 hours of free time, whereas the typical group reported having on average 14.25 hours of free time. Comparison of a community autism sample to a convenience sample of undergraduate students appears to potentially not be appropriate, given the vast differences in amount of free time reported. The models that were conducted included a measure of % of free time spent playing games, but did not account for the fact that the groups had such a huge disparity in the amount of time the could play video games, likely stemming from differences in their number of hours spent in school or work.

ASD = 0 participants with no free time reported; 25 participants (almost half the sample!) with 50 or more hours of free time.
TD = 11 participants with no free time reported; 4 participants with 50 or more hours of free time.

I ran some quick stats for an example of how the confound of free time differences could undermine the validity of the findings for the current study:

For the autism group alone (n = 59), the Pearson Correlation between total free time and reported video game addiction symptoms is r = .4031, p < .002. Free time also correlates significantly with average hours of video game played for individuals with autism, r = .51, p < .001. Thus, more free time for the autism group could explain both higher hours of video game play AND higher video game addiction scores. This suggests that total free time should be included in the models to control for total differences in free time per group, and also may overall undermine the validity of the current data due to lack of controlling for potential confounds and group differences.

Given points described above, the discussion (lines 298 thru 315) may be incorrect in the interpretation of the data if total amount of free time explains video game addiction scores more strongly than actual video game use for individuals with autism. This should be explored more fully by the authors. It is known that adults with autism (even those who do not have an intellectual disability) are often not having their service needs met and are more likely to have reduced employment and schooling rates once they leave high school (Roux et al., 2014; Shattuck et al., 2012; Taylor, Henninger, & Mailick, 2015; Taylor & Seltzer, 2011). Are individuals with autism who don’t have either full time school or full time employment getting bored being stuck at home all day with nothing to do? The current study could explore a more complete picture of the experiences of these adults with the available data, and doing so would greatly add to the literature more than condemning individuals with autism for playing video games during their disproportionately large amounts of free time compared to their TD peers.

·

Basic reporting

There are (two) points that need addressing here:

1) The manuscript would benefit from a more nuanced consideration of pathological video game use as a construct. While the authors do acknowledge that there is no broad consensus on it in the introduction, I think more could be said here. For instance, recent work by Przybylski (2016) suggests that IGD measures can be particularly susceptible to mischievous responding, and perhaps more relevant to the current study, there is emerging evidence that pathological game use is not particularly stable over time (Scharkow et al., 2014; Przybylski et al., in review).

2) The authors discuss concepts around excessive gaming in the introduction, but it would be useful to provide some context to this discussion - for example, they note that children with ASD play >2hrs per day. Does this vary across the week, and how does this compare with TD controls? For instance, there are some suggestions that the impact that gaming (as a subcategory of screen time) has on wellbeing is quadratic in nature, and varies according to the time of week (Przybylski and Weinstein, 2017).

Some consideration of these issues in the introduction and discussion would be helpful.

Experimental design

The authors take a suitably robust and appropriate approach to the design and analysis for the study, and should be commended for making their data and materials available. Just a couple of comments regarding the methods:

1) In section 2.1, it would be useful to include age data about the participant sample.

2) As per a point I make in section 1, the authors appear to have data on average time spent playing video games (a) on weekdays and (b) weekends. Given the suggestion that weekend play may have less of a negative impact than weekday play, it would be useful to include these two measures separately within the analysis.

3) Were there any data on the types of video games that the participant sample played? 'Video game' is a broad church, and there are some suggestions that specific types of content and play context may have an impact on some behaviours (e.g. Jin & Li, 2017). Perhaps this is beyond the remit of the current study, but it would be interesting to see whether adults with ASD are systematically playing different types of games, in different ways to TD controls.

Validity of the findings

The authors conduct a robust analysis here that is appropriate to the question being asked. I think that it was particularly interesting that percent of free time spent playing games came out in the preferred model over average time spent playing. I wonder whether this in part relates to the (untested) idea that people are generally poor at estimating the amount of time they play video games, so the absolute measure favours relatively poorly against a relative measure.

At any rate, the authors provide a clear and concise analysis of the findings, and appropriately acknowledge the limitations of the study.

---

## Round 0.2 · Minor Revisions

Dear Joseph and colleagues

Thank you for the comprehensive revision of your paper.

As you will see, Reviewer 1 has reviewed the paper again and is now happy that the concerns raised in her original review were, as you note in your response, the result of a miscommunication. The revisions you have made to the paper as well as the codebook on OSF have clarified these issues to avoid future misunderstanding. Reviewer 1 does suggest that removing the correlation data from below the diagonal in Table 2 would aid comprehension of Table 3, and I think this is a good idea.

Unfortunately, I'm going to be slightly annoying and ask for one further small amendment. Ordinarily I would not ask for changes that were not requested in the first round of revisions. However, in re-reading your paper, I noticed that some of the Bayes Factors were extremely high. For example, you state that "This model was strongly preferred to the null effects model by a factor of 6.4×10^11-to-1." I wondered how plausible these values were, so I consulted an expert in Bayesian statistics. His recommendation was that you should report the classic statistics (F or t values) alongside the Bayesian statistics. I apologise for not raising this earlier, but it should be very quick and straightforward to add this information.

Reviewer 1 ·

Basic reporting

The text of the manuscript overall is clear, the raw data is supplied, and the study conforms to PeerJ standards. The new inclusion of the codebook improves readers' ability to interpret the raw data, as the headings for the data were previously not labeled in an intuitive way. The topic is potentially of importance to the field, particularly to autism researchers.

Experimental design

The authors have now clarified their measure of FREE_TIME in the raw data via addition of a codebook and have clarified how this variable was assessed in the body of their paper. It was previously unclear that autistic participants were asked to calculate the percentage scores in their head, rather than reporting on total number of hours of free time. This clarification in the methodology, along with the addition of separate analyses for TD and ASD participants greatly improves this study and its interpretation. The addition of the analysis for motive to play games is also helpful for interpreting the results.

Validity of the findings

Table 2 and Table 3: In Table 2, it appears as though the numbers are the same above & below the diagonal, whereas in Table 3, they represent different groups. It may help to change the format of Table 2 to make this distinction clearer (e.g., delete repeat values in Table 2, if possible). It would also be helpful if positive correlations were flagged in some way, to aid the readers in their interpretations.

---

## Round 0.3 · accepted · Accept

Thank you for making the requested edits. I'm happy to accept the paper for publication.